# Debiasing Recommendation with Personal Popularity

## ABSTRACT

Global popularity (GP) bias is the phenomenon that popular items are recommended much more frequently than they should be, which goes against the goal of providing personalized recommendations and harms user experience and recommendation accuracy. Many methods have been proposed to reduce GP bias but they fail to notice the fundamental problem of GP, i.e., it considers popularity from a *global* perspective of *all users* and uses a single set of popular items, and thus cannot capture the interests of individual users. As such, we propose a user-aware version of item popularity named *personal popularity* (PP), which identifies different popular items for each user by considering the users that share similar interests. As PP models the preferences of individual users, it naturally helps to produce personalized recommendations and mitigate GP bias. To integrate PP into recommendation, we design a general *personal popularity aware counterfactual* (PPAC) framework, which adapts easily to existing recommendation models. In particular, PPAC recognizes that PP and GP have both direct and indirect effects on recommendations and controls direct effects with counterfactual inference techniques for unbiased recommendations. Experimental results show that PPAC consistently outperforms SOTA debiasing methods across different datasets and base models, and the improvement in NDCG is up to 61.9%. All codes and datasets are available at https://anonymous.4open.science/r/Pop-4760/.

ACM Reference Format:
Anonymous Author(s). 2023. Debiasing Recommendation with Personal Popularity. In *Proceddings of ACM Web Conference (WWW '24), May 13-17, 2024, Singapore*. ACM, New York, NY, USA, 9 pages. https://doi.org/xx.xxxx/xxxxxxx.xxxxxxx

## 1 INTRODUCTION

Recommender systems aim to provide personalized suggestions to users and are prevalent in domains such as e-commerce, media, and social networks. They typically analyze user behaviors (e.g., clicks and purchases) and predict a score for each user-item pair, which indicates the chance that the user interacts with the item, and suggest items with highest prediction scores to the user [19, 20, 32].

Recently, popularity bias (called *global popularity bias* in this paper) attracts interests [5, 12, 32], which means that recommendation models (e.g., MF [39] and LightGCN [20]) often suggest excessive items with high global popularity (GP) values to users [16, 28, 54]. In particular, the GP of an item is defined as the proportion of users that have interacted with the item in all users. An example

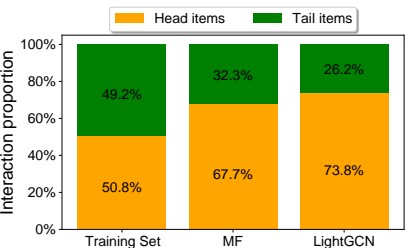

**Figure 1: Frequency of head and tail items appeared in the training set and recommendation lists of two well-trained models (i.e., MF and LightGCN) for MovieLens-1M dataset. Head items are the top 10% of items with the most user interactions in the training set, while tail items are the remaining.**

of GP bias is provided in Figure 1, where we train models on the MovieLens-1M dataset and count the frequency of head and tail items in the top-50 recommendation lists for all users. The results show that the head items (i.e., globally popular items) are recommended much more frequently than in the training set. GP bias leads to homogeneous recommendations to different users, which goes against the goal of providing personalized recommendations and is widely believed to harm user experience and recommendation accuracy [30, 38, 45]. Moreover, GP bias can cause the "Matthew Effect" – popular items are recommended to many users, and they become more popular and are suggested to even more users because of the frequent exposure to users. Thus, many "GP-aware" solutions have been proposed to mitigate GP bias (called *debiasing*) [16, 23, 46, 56].

**Personal popularity (PP).** Existing debiasing methods utilize the item GP but overlook the fact that GP is defined from a "global" perspective – it uses a single set of popular items (i.e., those with the largest GP values) for *all* the users. This implies that an item not interesting to a particular user may still be recommended due to its high GP, which harms user experience and recommendation accuracy. Moreover, different users may receive homogeneous recommendations because they share the same popular item set, which causes GP bias. To tackle the problems of GP, we propose a user-aware version of item popularity called *personal popularity (PP)*. In particular, for each user $u$, *PP* measures an item's popularity among the users that are similar to $u$, and we say that two users are similar if they have a large overlap in their historical interacted item sets. Compared with GP, PP considers the preferences of individual users and allows to identify a separate popular item set for each user. In Section 2, we show the benefits of PP using the MovieLens-1M dataset as an example. On the one hand, items with higher PP values tend to be rated better by users, which suggests that PP is informative and may help to improve the recommendation process. On the other hand, the personally popular items are different from the globally popular ones, and thus using PP can recommend items out of the globally popular item set and help to reduce GP bias.

**The PPAC framework.** To utilize PP for recommendation, we propose the *personal popularity aware counterfactual* (PPAC) framework with two design goals, i.e., (i) reduce GP bias of existing

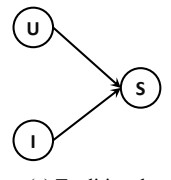
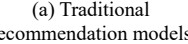

(a) Traditional
recommendation models

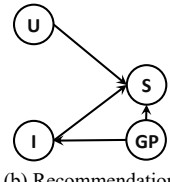

(b) Recommendation
with global popularity

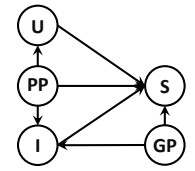

(c) Recommendation with
global and personal popularity

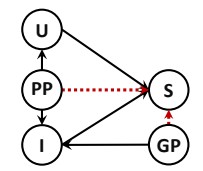

(d) Adjust PP → S and GP → S
by counterfactual inference

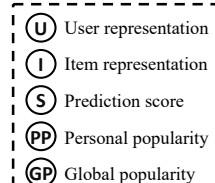

Ⓤ User representation
Ⓘ Item representation
Ⓢ Prediction score
㉠ Personal popularity
㉢ Global popularity

Figure 2: Causal graphs for existing methods (a-b) and our PPAC framework (c-d).

recommendation methods, and (2) be model-agnostic and easily support different base models (e.g., MF and LightGCN). To illustrate the differences between our PPAC and existing methods, we express them as "causal graphs" in Figure 2. Note that a causal graph is a directed acyclic graph (DAG) where the nodes represent variables, and directed edges indicate that one variable influences another.

Most recommendation models (e.g., MF and LightGCN) are designed based on the causal graph in Figure 2(a), which only utilizes the user and item representations for recommendations. Recently, some debiasing methods are developed according to Figure 2(b), which incorporates GP. However, we argue that in fact, Figure 2(c) models the real score generation process more accurately. Specifically, both PP and GP can directly impact the prediction score, as indicated by the paths $PP \rightarrow S$ and $GP \rightarrow S$. This is because users are more likely to know both globally popular items (e.g., due to the overall trends) and personally popular items (e.g., due to sharing among friends). Based on Figure 2(c), we propose the PPAC framework to consider the effects of both PP and GP in recommendation. To mitigate GP bias, PPAC framework estimates and adjusts the direct effects of GP and PP on the prediction score, as depicted by the red dotted lines in Figure 2(d). Since estimating the effects of the two paths is challenging, PPAC introduces a proxy variable [53, 60] to combine PP and GP using techniques from counterfactual inference. In particular, counterfactual inference imagines a counterfactual world where some variables are assigned to reference values [43, 49] and estimates how these variables affect the target variable.

We compare our PPAC with 10 baseline methods on 3 base models and 3 datasets. The results show that PPAC consistently outperforms all state-of-the-art debiasing baselines and the improvements over the best-performing baseline are up to 46.8% and 61.9% in terms of recall and NDCG, respectively. Moreover, experiment results also show that PPAC effectively reduces the recommendation frequency of globally popular items (i.e., mitigate GP bias) and suggest that PP benefits recommendation.

To summarize, we make the following contributions:

- To tackle the problem that existing GP cannot consider individual users, we propose a new definition of item popularity called PP, which captures the interests of individual users.
- To jointly consider PP and GP for recommendation debiasing, we design the PPAC framework using counterfactual inference techniques, which is model-agnostic.
- We conduct experiments to evaluate PPAC along with our designs and compare with state-of-the-art baselines.

## 2 PERSONAL POPULARITY

We consider the classical recommendation setting. For a given user set $\mathcal{U}$, an item set $\mathcal{I}$, and a set of user-item pairs $\mathcal{R}$ that record the

previous interactions of users (e.g., clicked, purchased) with items, the objective is to predict the likelihood of a user $u \in \mathcal{U}$ interacting with an item $i \in \mathcal{I}$. Recommendation models typically learn a score function $f(u, i) : \mathcal{U} \times \mathcal{I} \rightarrow \mathbb{R}$, where a higher score indicates a higher probability of interaction. Based on $f(u, i)$, items are ranked based on their scores in descending order, with the top-ranked ones suggested to $u$. Before defining our personal popularity (PP), we first recap the global popularity (GP) used in existing works [49, 54].

DEFINITION 1 (GLOBAL POPULARITY). *Given an item $i$, the global popularity (GP) is denoted by $g_i = |\mathcal{U}_i|/|\mathcal{U}|$, where $\mathcal{U}_i$ is the set of users who have interacted with $i$ before.*

Global popularity measures the attractiveness of an item among *all users* in the user set $\mathcal{U}$. As discussed before, GP fails to capture personal interests of individual users, which can lead to homogeneous recommendations and GP bias. To address this problem, we propose the PP to take user interests into account.

To calculate PP, we first define the *user similarity*.

DEFINITION 2 (USER SIMILARITY). *Given users $u$ and $v$, the user similarity, denoted by $sim_{u,v}$, is: $\frac{|\mathcal{I}_u \cap \mathcal{I}_v|}{|\mathcal{I}_u \cup \mathcal{I}_v|}$, where $\mathcal{I}_u$ and $\mathcal{I}_v$ are the sets of interacted items for user $u$ and $v$, respectively.*

Our user similarity is essentially the Jaccard similarity [26, 29], which measures the degree of overlap between two sets. We consider two users to be similar if they share similar interests, which is indicated by a large overlap in the items they have interacted with.

DEFINITION 3 (SIMILAR USER SET). *For a user $u$, the similar user set, denoted by $\mathcal{S}_u$, is the set of $k$ users that have the highest user similarity $sim_{u,v}$ with $u$. We denote a similar user as $v \in \mathcal{S}_u$.*

The number of similar users to consider, i.e., $k$, is a hyper-parameter.

DEFINITION 4 (PERSONAL POPULARITY). *Given a user $u$ and an item $i$, the personal popularity (PP), denoted by $p_{u,i}$, is:*

$$p_{u,i} = \frac{|\mathcal{S}_u^i|}{|\mathcal{S}_u|}, \tag{1}$$

*where $\mathcal{S}_u^i$ denotes the users in $\mathcal{S}_u$ that have interacted with item $i$.*

Note that $\mathcal{S}_u^i \subseteq \mathcal{S}_u$, and thus $p_{u,i} \in [0, 1]$. Conceptually, $p_{u,i}$ measures the attractiveness of item $i$ among a group of users that share similar interests with $u$. As such, $p_{u,i}$ reflects the personal interests of individual users and allows different users to have different popular item sets.

Our PP definition considers the basic collaborative filtering setting [20, 31, 39], which provides only user/item IDs and their interaction records. Such data are required by all recommenders, and thus our methods can be applied to any existing recommendation model. PP definition can be extended when side information is

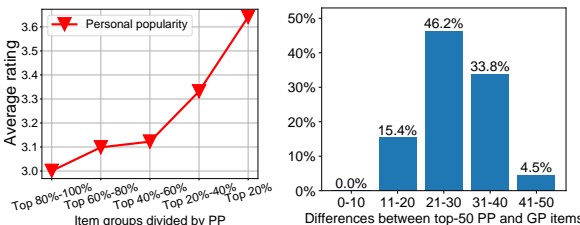

**(a) Rating over PP ranking**   **(b) User distribution on differences**

Figure 3: Analyzing personal popularity on MovieLens-1M.

available (e.g., interaction timestamps, user profiles, item descriptions), for instance, user profiles (e.g., age, gender, and geographics) allows to define more accurate user similarity and hence PP.

Besides, we argue that even though the current definition of PP is relatively simple, the idea of PP is generalizable that using a separate set of popular items for each user to capture user interests. Alternative definitions of PP are also possible, e.g., considering multi-hop connectivity between users and items. We leave the investigation of more sophisticated PP definitions as future work.

**Analyzing PP.** Given an item $i$, the GP value $g_i$ is the same for every user (Definition 1) but the PP value $p_{u,i}$ is user-dependent (Definition 4). As it is unlikely that two users have the same similar user set $S_u$ (Definition 3), there is a high chance that PP values vary among users. Moreover, PP addresses the interests of individual users, potentially making recommendations more personalized and reducing the recommendations of globally popular items.

We analyze PP using the *MovieLens-1M* dataset, which contains user ratings (on a scale of 1-5 stars) for movies. Table 1 gives the details of this dataset. We sorted all the items based on their average PP values, put them in five equal-size groups, and calculated the average user ratings for the items in each group. Figure 3(a) shows that items with high PP values tend to receive higher ratings from users, indicating PP can be useful for recommendation. We then examine whether a personally popular item is also globally popular. We extract the set $I_{GP}$ of items with the 50 highest GP values. For each user $u$, we also extract the set $I_{PP,u}$ of items with the 50 highest PP values. We then compute $d_u = |I_{PP,u} - I_{GP}|$, i.e., the number of items with 50 highest PP values that are not among the items with the top 50 GP values. Figure 3(b) shows the result grouped by users based on the $d_u$ values. We observe that (1) no user has $d_u$ less than 10; (2) $d_u > 20$ for around 85% of the users. These suggest that PP captures user-specific preferences that may not be adequately represented by GP.

## 3 PPAC FRAMEWORK

In this section, we first introduce the key concepts of counterfactual inference (Section 3.1) and then present our PPAC framework for debiasing from a causal view (Section 3.2). Next, we instantiate the PPAC framework with model designs (Section 3.3) and discuss model training and inference procedures (Section 3.4).

### 3.1 Preliminaries for Counterfactual Inference

To provide backgrounds for our PPAC framework, we introduce the basics of counterfactual inference [43, 53, 55], and more details can also be found in [36].

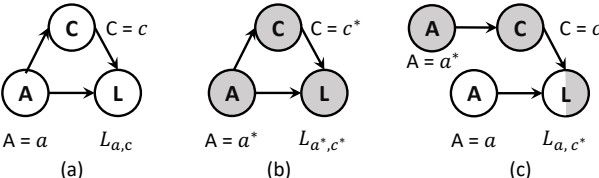

Figure 4: Causal graphs of the effect of alcohol consumption on lung cancer, where A, C, and L stand for alcohol, cigarette, and lung cancer. Gray nodes mean that the variables are at reference values (e.g., $A = a^*$).

**Causal Graph.** A causal graph is a directed acyclic graph (DAG), where nodes represent random variables and edges represent the causal-effect relations between variables. Figure 4(a) shows an example causal graph. There, we use a capital letter (e.g., $A$) to denote a random variable and a lowercase letter (e.g., $a$) to denote its observed value. $A \rightarrow L$ means that there is a *direct effect* of alcohol $A$ on lung cancer $L$. $A \rightarrow C \rightarrow L$ means that $A$ has an *indirect effect* on $L$ via $C$, which acts as a mediator [53] because people who drink alcohol may be more likely to smoke and hence tend to get lung cancer. The value of $L$ can be calculated from the values of its ancestor nodes and formulated as:

$$L_{a,c} = L(A = a, C = c), \quad c = C_a = C(A = a), \quad (2)$$

where $L(\cdot)$ and $C(\cdot)$ are the structural equations of $L$ and $C$, respectively. $C_a$ denotes the cigarette consumption of a person assuming that he/she has an alcohol consumption of $a$. $L_{a,c}$ is the lung cancer outcome of the person if he/she has alcohol consumption $a$ and cigarette consumption $c$. To calculate direct effect and indirect effects of $A$ on $L$ separately, we should use counterfactual inference.

**Counterfactual Inference.** Counterfactual inference is essentially a thinking activity that imagines the outcomes of changing the value of a single variable [22]. For the example in Figure 4(b), it considers "*what would happen if alcohol consumption was set to another value?*". Gray nodes mean that the variables are at a *reference state* (i.e., assigned a *reference value*, e.g., $A = a^*$), which is an intervention independent from the facts and used to estimate causal effects [54]. Figure 4(c) shows a causal graph of the counterfactual world where $C$ is set to $c^* = C(A = a^*)$ and $L$ is set to $L_{a,c^*} = L(A = a, C = c^*)$. Note that this is only an imagined scenario created to study the effect of $A$ on $L$. It is called a counterfactual scenario because we are combining the fact and assumption by setting both $A = a$ and $A = a^*$ simultaneously, even though it will not occur in reality [43].

**Causal Effect.** The causal effect of $A$ on $L$ is the extent to which the value of the target variable $L$ changes when an ancestor node $A$ experiences a unit change [49]. For instance in Figure 4, the *total effect* (TE) of $A = a$ on $L$ is defined as:

$$TE = L_{a,c} - L_{a^*,c^*}. \quad (3)$$

TE can be decomposed into the sum of *natural direct effect* (abbreviated as NDE, i.e., effect via path $A \rightarrow L$) and *total indirect effect* (abbreviated as TIE, i.e., effect via path $A \rightarrow C \rightarrow L$), which can be calculated as:

$$NDE = L_{a,c^*} - L_{a^*,c^*},$$
$$TIE = TE - NDE = L_{a,c} - L_{a,c^*}. \quad (4)$$

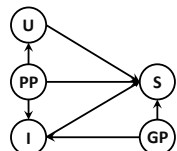 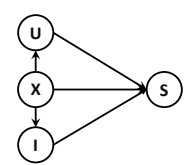 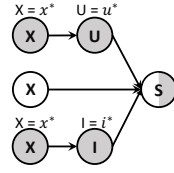

(a) Factual world     (b) Factual world (Proxy variable)     (c) Counterfactual world

**Figure 5: Causal graphs of GP and PP aware recommendation in factual and counterfactual worlds. In plot (b-c), X is a proxy variable that models the combination of GP and PP.**

## 3.2 Counterfactual Debiased Recommendation

In this section, we first discuss how GP and PP affect the prediction scores via a causal graph, and then introduce the rationale of our PPAC framework to achieve debiasing.

In Figure 5(a), we abstract the factual world recommendation into a causal graph, where $U, I, PP, GP, S$ are user representation, item representation, personal popularity, global popularity, and user-item prediction score, respectively. The prediction score $S$ models the probability of a user-item interaction. From the perspective of causality, both GP and PP can directly affect the prediction score via $PP \rightarrow S$ and $GP \rightarrow S$ because items with high GP and PP values are more likely to be known by the users and thus be recommended. We argue that adjusting the direct effect of these two paths is an effective way to achieve debiasing, since it can result in a more flexible recommendation.

It is very challenging to estimate the direct effects of $PP \rightarrow S$ and $GP \rightarrow S$ separately, and most of existing work handles causal graphs with only one direct effect path [43, 49, 54]. Inspired by proximal causal inference [53, 60], we introduce a new variable $X$ in Figure 5(b), which is a *proxy variable* that incorporates different confounders in the single-treatment regime. In particular, we use a single node $X$, which mixes PP and GP, to calculate the causal effects of them on $S$. Note that since PP can affect both $U$ and $I$, the proxy variable $X$ should also be able to affect both $U$ and $I$.

Now, the problem becomes how to estimate the direct effect of $X$ on $S$. Following the causal effect calculation introduced in Section 3.1, we first construct the counterfactual world in Figure 5(c). According to Eq. (3), the total effect (TE) of $X$ can be written as:

$$TE = S_{x,u,i} - S_{x^*,u^*,i^*}. \tag{5}$$

The indirect paths (i.e., $X \rightarrow U \rightarrow S$ and $X \rightarrow I \rightarrow S$) are blocked by setting $U$ and $I$ to reference states, i.e., $I = i^* = I(X = x^*)$ and $U = u^* = U(X = x^*)$. After that, we can calculate the natural direct effect (NDE) of $X$ as:

$$NDE = S_{x,u^*,i^*} - S_{x^*,u^*,i^*}. \tag{6}$$

According to Eq. (4), the total indirect effect (TIE) is

$$TIE = TE - NDE = S_{x,u,i} - S_{x,u^*,i^*}. \tag{7}$$

Recall that our goal is to control the direct effect of $X \rightarrow S$ (i.e., NDE) in recommendation to mitigate GP bias. As such, we obtain debiased predictions by counterfactual inference as follows:

$$\begin{aligned} TIE + \epsilon NDE &= S_{x,u,i} - S_{x,u^*,i^*} + \epsilon(S_{x,u^*,i^*} - S_{x^*,u^*,i^*}) \\ &= S_{x,u,i} - \epsilon S_{x^*,u^*,i^*} + (\epsilon - 1)S_{x,u^*,i^*}, \end{aligned} \tag{8}$$

where $\epsilon$ is a hyper-parameter that controls the weight of NDE. One might argue that reducing GP bias may require shrinking the GP effect and enlarging the PP effect from an intuitive view. However, we employ a proxy variable to estimate the NDE by combining GP and PP. It is important to note that the proxy variable is solely for helping to estimate the NDE. When we make adjustments to the NDE using models in practice, we will separately modify the GP and PP effects, which will be detailed in next section.

## 3.3 Model Designs

In this section, we introduce how to instantiate counterfactual debiased recommendation in Section 3.2 with model designs. According to Eq. (8), we need to estimate $S_{x,u,i}$, $\epsilon S_{x^*,u^*,i^*}$ and $(\epsilon - 1)S_{x,u^*,i^*}$.

**Estimate $S_{x,u,i}$.** Unlike traditional recommendation models that rely solely on user and item representations to estimate prediction scores, we explicitly consider the importance of $X$ in the estimation process. To achieve the estimation of $S_{x,u,i}$, we multiply [49] the estimated value of $X$ (denoted as $\hat{x}_{u,i}$) with the prediction score of the existing recommendation model $f_R$, which we refer to as the *base model* that needs to be debiased, as follows:

$$S_{x,u,i} = \hat{x}_{u,i} * f_R(U = u, I = i), \tag{9}$$

where $f_R$ can be any existing recommendation model that needs to be debiased. Recall that $X$ combines both GP and PP, so we implement $\hat{x}_{u,i} = \sigma(f_{PP}(U = u, I = i)) * \sigma(f_{GP}(I = i))$, where $\sigma$ is the sigmoid function and $\sigma(f_{PP}(\cdot))$ and $\sigma(f_{GP}(\cdot))$ are the estimation models of PP and GP, respectively. $f_{PP}$ and $f_{GP}$ are implemented as Multi-layer Perceptions (MLPs) for simplicity but they can also be replaced with any neural network. Since GP is an item-specific property and PP is a property of both users and items, $f_{GP}$ only takes items as input, while $f_{PP}$ takes both users and items as input. The sigmoid function ensures that the estimated GP and PP values fall within the (0, 1) range, preventing them from being equal to zero and invalidating the other models.

**Estimate $\epsilon S_{x^*,u^*,i^*}$.** In this context, all the variables are considered to be at reference states, implying that the causal effects of all paths in Figure 5(b) are fixed. Therefore, $S_{x^*,u^*,i^*}$ represents a constant value that is independent of the specific users and items involved. Regardless of the value of $\epsilon$, $\epsilon S_{x^*,u^*,i^*}$ remains consistent for all user-item predictions. Therefore, adding this term has no impact on the item rankings for a user, allowing us to disregard it directly.

**Estimate $(\epsilon - 1)S_{x,u^*,i^*}$.** Here, $u$ and $i$ are reference states, which means that $S$ is unaffected by paths $X \rightarrow U \rightarrow S$ and $X \rightarrow I \rightarrow S$. As explained in Section 3.1, both $X = x$ and $X = x^*$ can coexist in the counterfactual world. In our context, we estimate $S_{x,u^*,i^*} = \hat{x}_{u,i} * f_R(U = u^*, I = i^*)$, where $f_R(U = u^*, I = i^*)$ is a fixed constant value that blocks the effects of $U \rightarrow S$ and $I \rightarrow S$, while the value of $\hat{x}_{u,i}$ still depends on the specific user and item. Next, we set $\tau = (\epsilon - 1)f_R(U = u^*, I = i^*)$, which acts as an adjustable weight for $\hat{x}_{u,i}$, allowing us to adjust the influence of $X$ (i.e., the combination of GP and PP). For a better adjustment, we implement

$$\begin{aligned} (\epsilon - 1)S_{x,u^*,i^*} &= \tau \hat{x}_{u,i} \\ &= \gamma * p_{u,i} + \beta * g_i, \end{aligned} \tag{10}$$

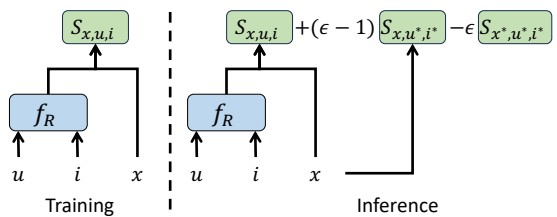

**Figure 6: Training and inference in PPAC framework.**

where $\gamma$ and $\beta$ are tunable weights for PP and GP, respectively. $g_i$ is the observed GP of item $i$ calculated on the training set, $p_{u,i}$ is the observed PP for user $u$ and item $i$ evaluated on the training set.

Note that we use the observed values of GP and PP in the estimation of $(\epsilon - 1)S_{x,u^*,i^*}$ while using model-predicted values in the $S_{x,u,i}$ estimation. This is because when user and item representations are at a reference state, they cannot directly affect the prediction scores. Nevertheless, $f_{GP}$ and $f_{PP}$ use user and item representations as inputs, and their parameters change continuously during the training process. Consequently, we replace the predicted values with the fixed observed ones when estimating $(\epsilon - 1)S_{x,u^*,i^*}$. Experiments in Section 4.3 show the effectiveness of this design.

### 3.4 Training and Inference

Next, we proceed to present our procedures for model training and inference. Figure 6 provides an overview of these processes.

**Training.** According to the above-mentioned estimation processes, we need to train $f_R$, $f_{PP}$ and $f_{GP}$.

To train $f_R$, note that the objective of training is to make the model predictions match the distribution of the training set, which is heavily biased, rather than conducting recommendations for users. Therefore, we use different methods to estimate the user-item interaction probability in training and inference phases. Specifically, since the training set is created via the causal graph in the factual world (Figure 5(a)) where all the causal effects have not been regulated. Therefore, to estimate the interaction probability $\hat{y}_{u,i}$ of a user $u$ and an item $i$ in the historical training set, we adhere to the estimation procedure of the prediction score $S_{x,u,i}$, which assumes that all paths in Figure 5(a) are unconstrained and can affect the prediction score. That is

$$\hat{y}_{u,i} = \hat{x}_{u,i} * f_R(U = u, I = i)$$
$$= \sigma(\hat{p}_{u,i}) * \sigma(\hat{g}_i) * \hat{r}_{u,i}. \tag{11}$$

Here, $\hat{r}_{u,i} = f_R(U = u, I = i)$ represents the prediction of the base model. $\sigma(\hat{p}_{u,i}) = \sigma(f_{PP}(U = u, I = i))$ and $\sigma(\hat{g}_i) = \sigma(f_{GP}(I = i))$ are the model-predicted PP and GP values, respectively. Then, we apply the Bayesian Personalized Ranking (BPR) loss [20, 33, 39] that is widely used in recommendations to train them:

$$L_R = \sum_{(u,i^+,i^-) \in O} -\ln \sigma(\hat{y}_{u,i^+} - \hat{y}_{u,i^-}), \tag{12}$$

where $O = \{(u, i^+, i^-)|(u, i^+) \in \mathcal{R}, (u, i^-) \in \mathcal{R}^-\}$ denotes the training set and $(u, i^+, i^-)$ is a training sample. Here, $\mathcal{R}$ is the set of observed user-item interactions, $i^+$ is a positive sample that $u$ has interacted. The set $\mathcal{R}^-$ contains randomly sampled unobserved user-item pairs; $i^-$ is a randomly sampled negative sample that $u$ has not interacted before.

**Table 1: Statistics of the experimental datasets.**

|  | MovieLens-1M | Gowalla | Yelp2018 |
|---|---|---|---|
| #User | 6,038 | 29,858 | 31,668 |
| #Item | 3,883 | 40,981 | 38,048 |
| #Interaction | 1,000,209 | 5,946,257 | 8,827,696 |

To train $f_{PP}$ and $f_{GP}$ (i.e., PP and GP estimation models), we regard them as regression tasks and use observed values as ground truth, then apply the Mean Squared Error (MSE) loss [40, 50]:

$$L_P = \frac{1}{|\mathcal{R}|} \sum_{(u,i) \in \mathcal{R}} (p_{u,i} - \sigma(\hat{p}_{u,i}))^2,$$
$$L_G = \frac{1}{|\mathcal{I}|} \sum_{i \in \mathcal{I}} (g_i - \sigma(\hat{g}_i))^2. \tag{13}$$

Putting them all together, the final loss function is

$$L = L_R + \alpha(L_P + L_G) + \lambda \|\Theta\|_2^2, \tag{14}$$

where $\alpha$ is a tunable hyper-parameter, $\Theta$ denotes all trainable parameters in the model, $\lambda$ is the weight of $L_2$ regularization.

**Inference.** Instead of relying on the predictions in the factual world, we adjust the NDE and apply Eq. (8) to make recommendations as explained in Section 3.2. By combining the previous estimations, we obtain the following inference formulation.

$$\sigma(\hat{p}_{u,i}) * \sigma(\hat{g}_i) * \hat{r}_{u,i} + \gamma * p_{u,i} + \beta * g_i. \tag{15}$$

Here, $\gamma$ and $\beta$ act as the adjustable weights for PP and GP, respectively. We also explain this counterfactual inference by an intuitive example. Consider two items $i$ and $j$ that are equally popular among all users (i.e., $g_i = g_j$) but $i$ is more popular than $j$ among users who share similar interests with user $u$ (i.e., $p_{u,i} > p_{u,j}$). If Eq. (11) estimates the ranking scores as $\hat{y}_{u,i} < \hat{y}_{u,j}$, $j$ will be ranked higher than $i$. By counterfactual inference, we can tune the $\gamma * p_{u,i}$ term, resulting in a more accurate ranking where $i$ is ahead of $j$. In our experiments, we observe that a positive value for $\gamma$ and a negative value for $\beta$ are effective in mitigating GP bias. These values amplify PP effects and decrease GP effects, making recommendation models combat GP bias better and predict user interests more precisely.

## 4 EXPERIMENTAL EVALUATION

In this section, we conduct extensive experiments to evaluate our PPAC framework and answer the following research questions:

- **RQ1:** Can PPAC outperform state-of-the-art debiasing methods?
- **RQ2:** How do the key designs of PPAC affect accuracy?
- **RQ3:** Does PPAC successfully mitigate global popularity bias?
- **RQ4:** How do different hyper-parameters affect PPAC?

### 4.1 Experiment Settings

**Datasets.** We conduct experiments on three publicly available datasets, i.e., MovieLens-1M[1], Gowalla[2], and Yelp2018[3], and their statistics are reported in Table 1. They cover three different recommendation applications, i.e., movie, location, and business.

**Base models.** To prove the versatility of PPAC on different base models, we experiment with three representative models, namely

---

[1]https://grouplens.org/datasets/movielens/
[2]https://snap.stanford.edu/data/loc-gowalla.html
[3]https://www.yelp.com/dataset

**Table 2: Overall performance. we mark the highest results in bold and the second highest with underline. Impr. is the improvement of PPAC over the best-performing existing baseline (NOT including our proposed MostPPop). "*" denotes a statistically significant improvement over the best-performing existing baseline at the significance level of 0.05 on the paired t-test.**

| | | | Base | MostPop | IPS | IPS_C | LapDQ | INRS | DICE | PDA | MACR | MostPPop | PPAC | Impr. |
|---|---|---|---|---|---|---|---|---|---|---|---|---|---|---|
| MovieLens-1M | BPRMF | Recall | 0.2967 | 0.1349 | 0.3077 | 0.3089 | 0.3149 | 0.3074 | 0.3060 | 0.2885 | 0.3058 | 0.3347 | **0.3789*** | 20.3% |
| | | NDCG | 0.1864 | 0.0751 | 0.1920 | 0.1922 | 0.1935 | 0.1902 | 0.1836 | 0.1688 | 0.1826 | 0.1929 | **0.2294*** | 18.6% |
| | NCF | Recall | 0.3366 | 0.1349 | 0.3381 | 0.3389 | 0.3391 | 0.3387 | 0.3362 | 0.3429 | 0.3446 | 0.3347 | **0.3921*** | 13.8% |
| | | NDCG | 0.1985 | 0.0751 | 0.1983 | 0.1998 | 0.2002 | 0.2003 | 0.1989 | 0.2049 | 0.2089 | 0.1929 | **0.2365*** | 13.2% |
| | LightGCN | Recall | 0.3757 | 0.1349 | 0.3759 | 0.3768 | 0.3807 | 0.3811 | 0.3816 | 0.3846 | 0.3867 | 0.3347 | **0.4056*** | 4.9% |
| | | NDCG | 0.2295 | 0.0751 | 0.2297 | 0.2301 | 0.2334 | 0.2339 | 0.2291 | 0.2333 | 0.2381 | 0.1929 | **0.2481*** | 4.2% |
| Gowalla | BPRMF | Recall | 0.1313 | 0.0023 | 0.1296 | 0.1308 | 0.1221 | 0.1223 | 0.1115 | 0.0985 | 0.1349 | 0.1461 | **0.1661*** | 23.1% |
| | | NDCG | 0.0480 | 0.0008 | 0.0468 | 0.0478 | 0.0443 | 0.0444 | 0.0398 | 0.0340 | 0.0518 | 0.0571 | **0.0675*** | 30.3% |
| | NCF | Recall | 0.1022 | 0.0023 | 0.1046 | 0.1062 | 0.0986 | 0.1000 | 0.1006 | 0.1086 | 0.1125 | 0.1461 | **0.1535*** | 46.8% |
| | | NDCG | 0.0384 | 0.0008 | 0.0397 | 0.0405 | 0.0364 | 0.0380 | 0.0380 | 0.0401 | 0.0430 | 0.0571 | **0.0643*** | 61.9% |
| | LightGCN | Recall | 0.1480 | 0.0023 | 0.1639 | 0.1641 | 0.1540 | 0.1572 | 0.1468 | 0.1536 | 0.1662 | 0.1461 | **0.1885*** | 13.4% |
| | | NDCG | 0.0544 | 0.0008 | 0.0607 | 0.0608 | 0.0575 | 0.0585 | 0.0536 | 0.0573 | 0.0654 | 0.0571 | **0.0780*** | 19.3% |
| Yelp2018 | BPRMF | Recall | 0.0634 | 0.0051 | 0.0712 | 0.0721 | 0.0712 | 0.0681 | 0.0664 | 0.0525 | 0.0764 | 0.0793 | **0.0885*** | 24.3% |
| | | NDCG | 0.0233 | 0.0017 | 0.0269 | 0.0272 | 0.0269 | 0.0263 | 0.0250 | 0.0190 | 0.0281 | 0.0314 | **0.0361*** | 34.2% |
| | NCF | Recall | 0.0662 | 0.0051 | 0.0652 | 0.0662 | 0.0639 | 0.0615 | 0.0621 | 0.0772 | 0.0853 | 0.0793 | **0.1024*** | 20.0% |
| | | NDCG | 0.0248 | 0.0017 | 0.0253 | 0.0263 | 0.0246 | 0.0232 | 0.0234 | 0.0310 | 0.0358 | 0.0314 | **0.0434*** | 21.2% |
| | LightGCN | Recall | 0.0852 | 0.0051 | 0.0868 | 0.0884 | 0.0879 | 0.0882 | 0.0867 | 0.0907 | 0.0884 | 0.0793 | **0.1031*** | 13.7% |
| | | NDCG | 0.0326 | 0.0017 | 0.0333 | 0.0346 | 0.0338 | 0.0337 | 0.0328 | 0.0358 | 0.0341 | 0.0314 | **0.0414*** | 15.6% |

matrix factorization (MF)-based model (**BPRMF** [39]), neural network-based model (**NCF** [21]), and graph-based model (**LightGCN** [20]).

- **BPRMF** [39] trains an MF model using the BPR loss.
- **NCF** [21] replaces the inner product in MF models with neural networks when computing user-item scores.
- **LightGCN** [20] performs graph convolution [25] on the user-item interaction graph without linear activation functions.

**Baselines.** For a comprehensive comparison, we use 10 baselines, i.e., the vanilla **Base** model, 2 ranking-based methods (**MostPop** [23], **MostPPop**), and 7 existing state-of-the-art debiasing methods, including 2 Inverse Propensity Score (IPS)-based methods (**IPS** [40], **IPS-C** [7]), 2 regularization-based methods (**LapDQ** [2], **INRS** [24]), 3 causal graph-based methods (**DICE** [58], **PDA** [54], **MACR** [49]).

- **MostPop** [35] directly ranks all items by their GP and recommends the top items. This method does not consider debiasing, and we use it to provide reference results.
- **MostPPop** directly recommends items with top PP for each user. This method is designed by us to show the effectiveness of PP. This is *our proposed method* to provide reference results.
- **IPS** [40] adds a weight to each item to adjust it score, and the weight is negatively correlated with its GP.
- **IPS-C** [7] applies max-capping to the weights to reduce variance.
- **LapDQ** [2] reorders items in the predicted recommendation lists to trade-off recommendation accuracy and tail item coverage.
- **INRS** [24] proposes a constraint on the recommendation of globally popular items to enhance recommendation diversity.
- **DICE** [58] disentangles user interests and user conformity by splitting the user embeddings into two different embeddings that learn user interests and conformity, respectively.
- **PDA** [54] predicts user-item interaction using both user-item matching (i.e., the base model) and global popularity.
- **MACR** [49] disentangles global popularity and user conformity to handle GP bias.

**Evaluation method.** Note that the conventional evaluation strategy that randomly selects a sub-dataset to test cannot reflect the users' true preference distribution [49], since test data still exhibits severe GP bias. To measure the debiasing capabilities of models, we conduct experiments on an intervened test set following existing methods [6, 49, 58]. Specifically, we sample 10% interactions as the test set from the dataset in a manner that ensures all items receive an equal number of interactions, and another 10% as the validation set using the same way. The remaining interactions are used for training. By doing so, we create a counterfactual environment where the influence of GP bias is eliminated, allowing it to better reflect user preferences. We adopt the all-ranking protocol [20, 49, 54] and report two commonly used metrics, i.e., Recall [33] and NDCG [56]. Here, the performance is computed based on the top 50 results for each metric (i.e., Recall@50 and NDCG@50).

**Implementation details.** We implement all models using PyTorch and set the dimension of both user and item embeddings as 64. We tune all hyper-parameters by a grid search to obtain the best results. Specifically, we configure the number of graph convolution layers to 3 for graph-based models. The learning rate is 0.01, the training batch size is 8092 for all experiments. By default in PPAC, PP coefficient $\gamma$ is 256 and GP coefficient $\beta$ is -128. The regression loss weight $\alpha$ is 0.1 and $L_2$ regularization coefficient $\lambda$ is 1e-4. The number of similar users to consider for PP calculation (i.e., $k$) is 30.

## 4.2 Main Results (RQ1)

Table 2 shows the performance of PPAC compared with different baselines on different base models in terms of Recall@50 and NDCG@50. Key observations are as follows.

- PPAC consistently outperforms all baselines across various metrics, datasets, and base models. Compared to the best-performing existing debiasing method, improvements obtained by PPAC are statistically significant, with gains of up to 46.8% and 61.9% in terms of Recall and NDCG, respectively. It demonstrates the effectiveness of PPAC across different base models and datasets.
- Our proposed MostPPop is usually ranked second when BPRMF is the base model, suggesting that PP is very powerful and effective for recommendations. Moreover, MostPPop (i.e., recommend items with higher PP) consistently outperforms MostPop (i.e.,

**Table 3: Effect of different model components on PPAC.**

| | | MovieLens-1M | | Gowalla | | Yelp2018 | |
|---|---|---|---|---|---|---|---|
| | | Recall | NDCG | Recall | NDCG | Recall | NDCG |
| BPRMF | Base | 0.2967 | 0.1864 | 0.1313 | 0.0480 | 0.0634 | 0.0233 |
| | w/o CI | 0.3459 | 0.2089 | 0.1394 | 0.0495 | 0.0706 | 0.0270 |
| | w/o PP | 0.3525 | 0.2129 | 0.1139 | 0.0429 | 0.0641 | 0.0250 |
| | w/o GP | 0.3718 | 0.2251 | 0.1518 | 0.0587 | 0.0814 | 0.0323 |
| | PPAC | **0.3789** | **0.2294** | **0.1661** | **0.0675** | **0.0885** | **0.0361** |
| NCF | Base | 0.3366 | 0.1985 | 0.1022 | 0.0384 | 0.0662 | 0.0248 |
| | w/o CI | 0.3425 | 0.1998 | 0.1135 | 0.0442 | 0.0745 | 0.0285 |
| | w/o PP | 0.3424 | 0.2041 | 0.0901 | 0.0415 | 0.0804 | 0.0391 |
| | w/o GP | 0.3755 | 0.2254 | 0.1141 | 0.0542 | 0.0869 | 0.0527 |
| | PPAC | **0.3921** | **0.2365** | **0.1535** | **0.0643** | **0.1024** | **0.0634** |
| LightGCN | Base | 0.3757 | 0.2295 | 0.1480 | 0.0544 | 0.0852 | 0.0326 |
| | w/o CI | 0.3877 | 0.2378 | 0.1642 | 0.0614 | 0.0931 | 0.0357 |
| | w/o PP | 0.3840 | 0.2312 | 0.1604 | 0.0632 | 0.0905 | 0.0346 |
| | w/o GP | 0.3428 | 0.1951 | 0.1660 | 0.0635 | 0.0910 | 0.0353 |
| | PPAC | **0.4056** | **0.2481** | **0.1885** | **0.0780** | **0.1031** | **0.0414** |

recommend items with higher GP) by a large margin in all cases, showing that PP is more potent than GP.

- The improvement of PPAC is usually more significant for BPRMF and NCF than LightGCN. This is because LightGCN achieves the highest accuracy among the base models, and thus the room for improvement is smaller. Considering different datasets, the improvement of PPAC is smaller on MovieLens-1M than Gowalla and Yelp2018. This is because MovieLens-1M contains fewer interactions and thus is easier to learn for the models.

## 4.3 Ablation Study (RQ2)

**Effect of different components.** To gain deeper insights into the design of PPAC, we conduct ablation studies by disabling specific components. Specifically, we develop three variants:

- **w/o CI** disables counterfactual inference and uses the factual-world predictions for recommendations (i.e., Eq. (11)).
- **w/o PP** removes the PP estimation model (i.e., $f_{PP}$) and all the terms about PP during training and inference.
- **w/o GP** removes the GP estimation model (i.e., $f_{GP}$) and all the terms about GP during training and inference.

We compare these variants with the vanilla *base* model and PPAC. The results are reported in Table 3 and several observations can be made. (1) PPAC consistently performs better than all variants, indicating the correctness and effectiveness of all of our estimations and designs in debiasing recommendations. (2) *w/o CI* outperforms the base model in all cases, highlighting the necessity of our counterfactual inference, which can control the direct effects of PP and GP in recommendation effectively. (3) *w/o PP* usually leads to worse performance compared with *w/o GP* (only a few exceptions for LightGCN on MovieLens-1M — a strong model on a small dataset), indicating that PP is more potent than GP in debiasing and removing PP will result in a large performance drop.

**Effect of observed and predicted popularity.** Recall that in the estimation of $S_{x,u,i}$ and $(\epsilon - 1)S_{x,u^*,i^*}$ in Section 3.3, we utilize the predicted values of PP and GP in $S_{x,u,i}$ but the observed ones in $(\epsilon - 1)S_{x,u^*,i^*}$. Besides the theoretical explanations in Section 3.3, we also conduct experiments to showcase the effectiveness of this design. Specifically, we compare the performance of using only model-predicted popularity (i.e., $\sigma(\hat{p}_{u,i})$ and $\sigma(\hat{g}_i)$) or observed popularity (i.e., $p_{u,i}$ and $g_i$) in both estimations and modify the training and

**Table 4: Effect of predicted and observed values on PPAC.**

| | | MovieLens-1M | | Gowalla | | Yelp2018 | |
|---|---|---|---|---|---|---|---|
| | | Recall | NDCG | Recall | NDCG | Recall | NDCG |
| BPRMF | PPAC-Pred | 0.2997 | 0.1792 | 0.0985 | 0.0371 | 0.0573 | 0.0222 |
| | PPAC-Obs | 0.3472 | 0.2123 | 0.1585 | 0.0652 | 0.0798 | 0.0331 |
| | PPAC | **0.3789** | **0.2294** | **0.1661** | **0.0675** | **0.0885** | **0.0361** |
| NCF | PPAC-Pred | 0.3454 | 0.2088 | 0.0891 | 0.0310 | 0.0653 | 0.0253 |
| | PPAC-Obs | 0.3459 | 0.1967 | 0.1471 | 0.0571 | 0.0674 | 0.0247 |
| | PPAC | **0.3921** | **0.2365** | **0.1535** | **0.0643** | **0.1024** | **0.0634** |
| LightGCN | PPAC-Pred | 0.2847 | 0.1518 | 0.1504 | 0.0558 | 0.0831 | 0.0324 |
| | PPAC-Obs | 0.3479 | 0.2114 | 0.1554 | 0.0665 | 0.0701 | 0.0304 |
| | PPAC | **0.4056** | **0.2481** | **0.1885** | **0.0780** | **0.1031** | **0.0414** |

inference steps correspondingly. Due to space constraints, we only report the results on the Gowalla dataset in Table 4. *PPAC-Pred* represents only using model-predicted GP and PP values while *PPAC-Obs* denotes only using observed ones.

For the results, we find that PPAC outperforms both variants in all cases, indicating that our estimation procedures are correct and effective. From an intuitive view, the observed popularity is more accurate and hence more suitable for fine-tuning popularity effects, while predicting popularity plays a crucial role in learning better user/item embeddings, resulting in more accurate predictions.

## 4.4 Model Analysis

**Debiasing Analysis (RQ3).** To comprehend the debiasing ability of PPAC, we divide the items into different groups based on the number of interactions they received in the training set, and Figure 7 reports the average recommendation frequency and accuracy (i.e., recall) for different item groups. The blue bars show the item proportion in each item group and the lines show the trends of frequency/recall of different baseline methods. Due to the limited space, we present the results on the Gowalla dataset. Note that the observations are similar for other datasets. We also include MACR for comparison as it usually ranks second among all existing baselines. From the results, we have the following observations.

- PPAC shows a reduced frequency of recommending items from the most globally popular item group compared to base models and MACR, as depicted in Figure 7(a-b). This suggests that PPAC is more effective in mitigating the influence of GP bias. Furthermore, Figure 7(c-d) illustrates that PPAC achieves the highest recall across all item groups, indicating its ability to better match users' true preferences and make more accurate recommendations compared to other baseline methods.
- The least globally popular item group (items with 0-10 interactions) experiences the largest increase in recall and receives more recommendations. This implies that traditional recommendation models are susceptible to GP bias and tend to recommend more items that are already globally popular. However, PPAC effectively avoids recommending items solely based on their high GP and instead focuses on recommending items to relevant users, thereby enhancing the recommendation of long-tail items.

**Hyper-parameters (RQ4).** We investigate how hyper-parameters (i.e., $\gamma$, $\beta$, and $k$) affect recommendations. Due to the limited space, we only present the results based on BPRMF model and Gowalla dataset but the results are similar under other configurations.

In Figures 8(a-b), we report recall@50 under different PP and GP coefficients $\gamma$ and $\beta$. Note that when adjusting $\gamma$ (resp. $\beta$), we keep

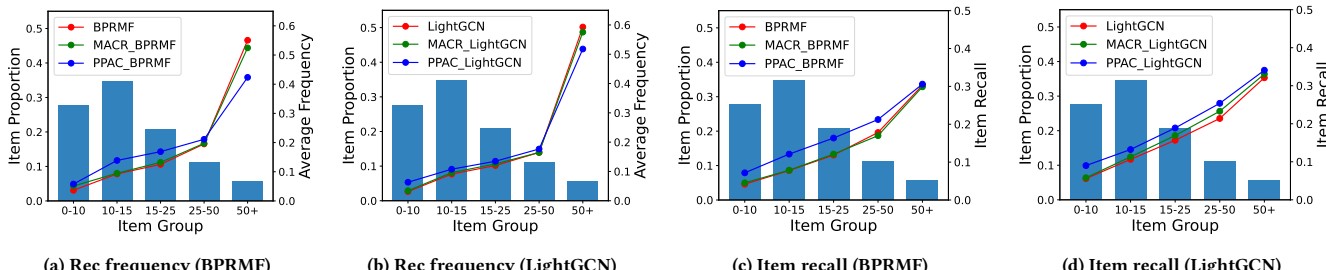

**Figure 7: Recommendation frequency and recall for different groups of items on Gowalla dataset.**

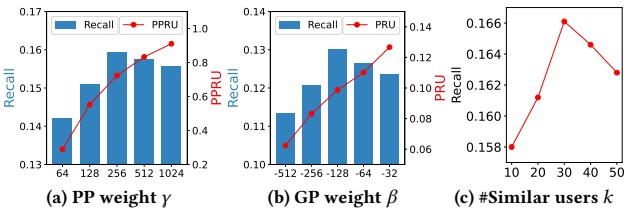

**Figure 8: Effect of hyper-parameters on Gowalla dataset.**

$\beta$ (resp. $\gamma$) 0 in this experiment, which is different from the joint tuning in other experiments. Here, we also report PRU@K [56, 61] and PPRU@K, which measure how well the item rankings according to the model-predicted scores agree with the item rankings according to GP and PP, respectively. The results show that recall first increases but then decreases for both coefficients, suggesting that it is essential to adjust the effect of GP and PP by counterfactual inference as in Eq. (15). Moreover, both PRU@K and PPRU@K increase with the corresponding popularity weight, showing that our counterfactual inference is effective—the final item ranking agrees better with the popularity ranking under a larger weight.

We plot the number of similar users $k$ used to compute PP against recall in Figure 8(c). The results show that there exists an optimal $k$, and switching to either smaller or larger values degrades accuracy. This is because using a large $k$ will include users who are not similar to the target user while a small $k$ means that not all users with similar interests are considered.

## 5 RELATED WORK

**Popularity bias in recommendation.** Due to the feedback loop of recommender systems (i.e., exposure affects interaction) [9], bias will be amplified and get increasingly serious. Several methods [4, 18, 27, 41, 51] are proposed to mitigate the global popularity bias since it has been proven to hinder user exploration and drive the users to homogenization [1, 3, 8, 10]. These methods can be classified into three categories. (1) *Inverse propensity score (IPS)-based methods*: IPS [40] weighs each interaction record using the reciprocal of the item's GP such that the interactions of globally popular items have a smaller influence on training. IPS-CN [18] adds weight normalization to IPS-C but introduces additional bias. (2) *Ranking adjustment or regularization*: LapDQ [2] re-ranks items in the predicted recommendation list to trade-off between accuracy and globally unpopular item coverage. ESAM [13] leverages regularization to transfer the knowledge learned from globally

popular items to globally unpopular items to tackle the problem that globally unpopular items do not have sufficient interactions. r-Adj [56] controls the normalization strength in the neighbor aggregation process of graph neural network [25, 44]-based models. (3) *Causal methods*: PDA [54] adopts the causal graph model and considers GP when computing the ranking scores. DICE [58] splits the user embeddings into two different embeddings that represent user interests and conformity, respectively, and learns user interests without the impact of user conformity. MACR [49] performs multi-task learning to estimate user interests, user conformity, and GP. TIDE [57] considers the change in GP over time. All these methods only consider GP, which applies a single set of popular items to all users and thus fails to model the interests of individual users. In contrast, our PP considers the interests of individual users by using a separate set of popular items for each user, which naturally combats GP bias and yields better recommendations.

**Causal recommendation.** The causal inference techniques have found applications in various fields, including computer vision [11, 34, 42], natural language processing [15, 17, 37], and information retrieval [6, 14, 52]. In recommender systems, causal inference can help understand the inherent causal mechanism of user behavior [55]. For instance, CR [47] addresses the click-bait issue [47] by intervening in the exposure effects. COR [48] handles user feature shifts and out-of-distribution recommendations by controlling the user features. HCR [59] adopts a front-door adjustment-based method to decompose the causal effects of user feedback and item features. MCMO [60] models recommendation as a multi-cause multi-outcome inference problem and handles exposure bias. We focus on the GP bias, which is a different problem from the aforementioned works, and develop a causal graph that incorporates both PP and GP to improve recommendation models.

## 6 CONCLUSIONS

This paper proposes a new notion of item popularity termed personal popularity (PP), which finds different sets of popular items for each user by considering individual user's interests, while existing global popularity (GP) only finds a single set of popular items for all users. Therefore, PP can naturally mitigate GP bias and conduct better recommendations. Furthermore, we propose the Personal Popularity Aware Counterfactual (PPAC) framework, which achieves debiasing by incorporating PP and GP and controls their direct effects on recommendations using counterfactual inference techniques. Extensive experiments show that PPAC significantly outperforms existing recommendation debiasing methods.

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
