# OpenReview forum: "Debiasing Recommendation with Personal Popularity"
_ACM.org/TheWebConf/2024/Conference — TheWebConf24 Oral_

### Official Review · Reviewer_GWGk · 2023-11-16

**Novelty:** 4
**Technical Quality:** 4

**Review:**

This study focuses on addressing the prevalent issue of global popularity bias within recommendation systems. It introduces the concept of personal popularity and integrates two types of popularity to enhance recommendation accuracy. Comprehensive experiments have been conducted to illustrate the superiority, feasibility, and validity of the proposed framework. The provision of code and datasets further enhances the transparency and reproducibility of the research, contributing to the overall robustness of the findings. The paper is easy to follow.

However, given the widespread popularity of recommendation as a research topic with a continual influx of related studies, it's noteworthy that debiasing, in this context, focuses on addressing just one facet to enhance recommendation accuracy. Thus, the efficacy of the proposed method should also be substantiated through comparisons with other contemporary recommendation methods, and not limited to debiasing based recommendation methods. This would further elucidate the superiority of the proposed approach.

While numerous relevant studies have emerged over the past three years, and the authors have addressed some, such as [27, 32, 23], along with various studies published in 2022 and 2023, it is noteworthy that the evaluation only compares the proposed framework to competitive methods dating back to 2021. To underscore the significant contribution of the proposed framework, it is recommended to include comparisons with more recent studies in the evaluation.

**Questions:**

Demonstrate the superiority of the proposed framework over other SOTA methods.

**Ethics Review Description:**

Nothing

**Reviewer Confidence:**

4: The reviewer is certain that the evaluation is correct and very familiar with the relevant literature

**Scope:**

4: The work is relevant to the Web and to the track, and is of broad interest to the community

---

### Official Review · Reviewer_Jngt · 2023-11-17

**Novelty:** 4
**Technical Quality:** 5

**Review:**

In this paper, the authors proposed PP-personal popularity and considered both PP and GP (global popularity) in recommending items. Though it is a very simple concept, the authors claimed that PP is important and can benefit the recommendation. The authors designed a model to estimate the causal effect and obtained the debiased predictions via the model.

Pros:
- The paper is well-written and easy to follow.
- The authors made comprehensive ablation to show the effectiveness of the model design.

Cons:
 - The PP is not a novel concept, according to Eq.(1), it measures the number of interactions among similar users. Actually, in my opinion, the idea of PP is very similar to the basic idea used in KNN. Can the author further clarify the difference here?

- The objective of the work could be stated more clearly, i.e., what exactly is the question the authors want to solve, and what is the objective (e.g., higher accuracy? less GP? less PP?)

**Questions:**

1. see the first question in Cons. Besides, in the experiment results, I noticed that MostPPop (directly recommending items with top PP) outperforms not only MostPop but also a great number of baselines. This phenomenon is interesting. Can authors provide more insights regarding this?

2. It appears that the model is designed to mitigate the effect of GP, but increase the effect of PP (If I am incorrect, please inform me). Could the authors provide an explanation of why this approach would be effective? Because the GP and PP are quite similar in many ways, the only difference is that one is defined globally and one is defined locally.

3. In the "evaluation method" of Section 4.1 they said they sampled items that received the same interactions in the test set to "create a counterfactual environment where the influence of GP bias is eliminated, allowing it to better reflect user preferences". However, I do not understand why such a setting is able to measure the debiasing capabilities of models. An intuitive way is to count and compare the GP in the recommended top K item lists. Can authors give more explanations regarding this?

**Reviewer Confidence:**

4: The reviewer is certain that the evaluation is correct and very familiar with the relevant literature

**Scope:**

4: The work is relevant to the Web and to the track, and is of broad interest to the community

---

### Official Review · Reviewer_1LVH · 2023-11-20

**Novelty:** 6
**Technical Quality:** 6

**Review:**

Summary
This work proposes to substitute the commonly used global definition of item popularity with a local definition of item popularity, i.e., the (personalized) item popularity is measured based on the popularity of an item within a target user’s neighborhood. Via integrating this definition into a counterfactual popularity bias mitigation framework, the authors show that their proposed metric is superior to global popularity metrics.

Strengths
S1: The propose personalized popularity bias metric is conceptually simple, however, I think this is obvious that measuring unpersonalized popularity does not reflect a user’s perception of item popularity well. Thus, it makes total sense to use personalized popularity.
S2: The concept behind their approach is well described. Many details on their approach are provided. Here I’d like to highlight several Figures (Figures 2, 4, 5, 6) that help the reader to understand the approach.
S3: The experiments compare the proposed approach to multiple appropriate baselines and use several different datasets. The authors test the accuracy, and also, perform an ablation study to pinpoint the impact of single components on the superiority of their approach. Also the impact of hyperparameters is tested.

Weaknesses/questions - see below

**Questions:**

1: The experiments in regard to RQ3 (How does the approach impact popularity bias?) are limited. If I am not mistaken, Figure 7 is the only plot/result that studies popularity bias. Since popularity bias mitigation is a strong contribution of this work, I recommend to focus more on popularity bias experiments (e.g., coverage, calibration, compare items w.r.t. their personalized popularity vs. global popularity, …). Could you please elaborate more on that?

2: Some (minor) points weaken the quality of this work. I think Figure 1 is obvious and not needed. I recommend to rather visualize the difference between personalized and global popularity (e.g., via plotting the recommendation network). On Page 3 (Analyzing PP), I believe the notation |I_PPu – I_GP| is wrong. Should it be | PPu | - |I_GP|? In Section 3.2.: What does x*, u*, and i* stand for? This is not defined in the text. In Section 4.2., what statistical test is used? Also, from the caption in Table 3 it is not clear what values are bold or underlined. On Page 8 (Hyper-parameters), the authors refer to the metrics PRU and PRRU. However, for the reader, it is poorly (not) explained what these metrics stand for and how they are defined. The authors’ explanation is not sufficient.

3: I think the authors should think about pulling the Related Work to the beginning of the paper (currenty it’s at the end), since it contains many points that could help to fit this paper’s contributions into the research landscape. What do you think?

**Ethics Review Description:**

-

**Reviewer Confidence:**

3: The reviewer is confident but not certain that the evaluation is correct

**Scope:**

4: The work is relevant to the Web and to the track, and is of broad interest to the community

---

### Official Review · Reviewer_4Tt6 · 2023-11-24

**Novelty:** 4
**Technical Quality:** 4

**Review:**

This work proposes user-aware item popularity (personal popularity) ,by users’ similarity and similar user set to identify different popular items for each user, to address the problem of popular items being frequently recommend. It helps generate personalized recommendations and alleviate the bias of global popularity, by modeling personal popularity based on individual and her similar users’ preferences. To incorporate personal popularity into the recommendation process, this paper designs a Personal Popularity Aware Counterfactual (PPAC) framework to identify the direct and indirect impacts of personal popularity and global popularity on recommendation.

Pros:
1. The structure of the paper is clear. The data analysis and demonstration of the problem are well-executed.
2. The description of the problem to be solved and the proposed solution is very clear.
Cons:
1. The experiments are not sufficient.  (1)The model proposed in this paper is only combined with three basic models for experiments, which makes the results inconclusive. (2) Not integrated with the latest recommendation model in experiments. (3 ) The metrics for evaluating experiment performance are also few. (4) The performance comparison in this paper is only based on the top-50 recommended results. The experimental results will be more convincing if you add more different top-k recommendation comparison experiments.

**Questions:**

1. In Definition 2, user similarity is represented by the overlap of the interactive items of two users. However,users’ interaction may include clicking, collecting, purchasing, etc., and different interactive behaviors represent different preferences degree of users.  In the paper,Which interactive behaviors are used to calculate the user similarity?
2. In this paper, K users with high similarity are selected to form the similar user set of the target user. When the number of similar users is less than k, how does the paper deal with it? Why not consider setting a similarity threshold where users greater than this threshold are similar users of the target user?
3. For cold start users, they have few similar users or without similar users, does the paper consider how to recommend?
4. The Contribution 1：“We propose a new definition of item popularity called PP........，”, it should be “personal popularity”?
5. The contributions in the paper should be described in detail.

**Ethics Review Description:**

Non

**Reviewer Confidence:**

4: The reviewer is certain that the evaluation is correct and very familiar with the relevant literature

**Scope:**

4: The work is relevant to the Web and to the track, and is of broad interest to the community

---

### Official Review · Reviewer_y2gV · 2023-11-29

**Novelty:** 6
**Technical Quality:** 6

**Review:**

The paper introduces a framework termed Personal Popularity Aware Counterfactual (PPAC) to debias recommender systems. The paper distinguishes between global and personal popularity, whereas the latter corresponds to a user-aware version of item popularity. The framework leverages counterfactual inference to debias recommendations with respect to popularity. Experiments are conducted on several datasets and the approach is compared with other state-of-the-art debiasing techniques.

Pros:
- The paper presents an original approach to addressing popularity bias in recommender systems. The concept of personal popularity is innovative, and its integration into recommendation systems using counterfactual inference is a significant contribution.
- The consideration of both personal and global popularity is interesting.
- The paper is very well written and clearly structured.
- Extensive experiments are presented, which support the paper's main research goals.
- Parameter analysis is performed and an ablation study sheds light on the influence of particular components on the performance of the approach.

Suggestions for improvement:
- it would be interesting to see how this approach interacts with other types of biases in recommender systems, or to have at least a brief discussion on this in the paper.´
- It is not clear how the framework deals with cold-start users, this should be explained.
- The analysis of the debiasing performance is rather short. Given that debiasing of popularity bias is one of the selling points of the paper, more details and also experiments would be helpful.
- Equations 5, 6, 7, 8 are not well described and lack a description of the variables. This makes it hard to correctly interpret them.
- Reproducibility is limited since the code for the experiments is apparently not published.

**Questions:**

Could you give more details and insights of the debiasing experiments?
Please explain equations 5-8 in greater detail (see comment above).
Will you release the code, so others can reproduce your work?

**Ethics Review Description:**

No ethics issues found

**Reviewer Confidence:**

4: The reviewer is certain that the evaluation is correct and very familiar with the relevant literature

**Scope:**

4: The work is relevant to the Web and to the track, and is of broad interest to the community

---

### Decision · Program_Chairs · 2024-01-22

**Decision:**

Accept (Oral)

**Comment:**

Quality:
 + Innovative approach to mitigate popularity bias in recommender systems.
 + Results of extensive experiments are reported.
 + Authors provide link to code and data repository.
 + Authors provide detailed responses to issues and questions raised by reviewers.
 - Some structural concerns (e.g., placement of Related Work)
 - Some concerns about simplicity and novelty (which the AC does not share).
 - Some most recent SOTA methods are not used as baselines.

 Clarity:
 + Overall, good.
 - Several equations are not well described, though.

 Originality:
 + The proposed PP bias metric is conceptually simple, nevertheless seems novel.

 Significance:
 + The work is highly relevant and of interest to the recommender systems community.